# Patient–Therapist Cooperative Hand Telerehabilitation through a Novel Framework Involving the Virtual Glove System

**DOI:** 10.3390/s23073463

**Published:** 2023-03-25

**Authors:** Giuseppe Placidi, Alessandro Di Matteo, Daniele Lozzi, Matteo Polsinelli, Eleni Theodoridou

**Affiliations:** 1A^2^VI-Lab, Department of Life, Health & Environmental Sciences, University of L’Aquila, 67100 L’Aquila, Italy; 2A^2^VI-Lab, Department of Information Engineering, Computer Science and Mathematics, University of L’Aquila, 67100 L’Aquila, Italy; 3Department of Computer Science, University of Salerno, 84084 Fisciano, Italy

**Keywords:** virtual glove, hand rehabilitation, rehabilitation framework, rehabilitation pipeline, hand tracking, telerehabilitation, stroke, hand surgery

## Abstract

Telerehabilitation is important for post-stroke or post-surgery rehabilitation because the tasks it uses are reproducible. When combined with assistive technologies, such as robots, virtual reality, tracking systems, or a combination of them, it can also allow the recording of a patient’s progression and rehabilitation monitoring, along with an objective evaluation. In this paper, we present the structure, from actors and functionalities to software and hardware views, of a novel framework that allows cooperation between patients and therapists. The system uses a computer-vision-based system named virtual glove for real-time hand tracking (40 fps), which is translated into a light and precise system. The novelty of this work lies in the fact that it gives the therapist quantitative, not only qualitative, information about the hand’s mobility, for every hand joint separately, while at the same time providing control of the result of the rehabilitation by also quantitatively monitoring the progress of the hand mobility. Finally, it also offers a strategy for patient–therapist interaction and therapist–therapist data sharing.

## 1. Introduction

Hand rehabilitation is crucial in the recovery of people with post-stroke or post-surgery deficits. Rehabilitation is usually performed face-to-face, by one therapist, for every treated person (one-to-one), or by several therapists working with one patient (many-to-one), especially when “many” refers to different times (different specialists operating in turn to the same patient) [1]. Therapists subjectively evaluate the efficacy of the procedure based on their experience. Recent studies suggest that repetitive and long-term training is beneficial for post-stroke people [1,2,3]. While the significance of face-to-face rehabilitation is indisputable, subjectively evaluated rehabilitation lacks precision for recording patient progression and/or for monitoring the rehabilitation at home [4]. This could mean that the progression is not properly evaluated or is evaluated differently by different therapists. Additionally, the rehabilitation program could differ from one therapist to the other, thereby producing different effects: therapy could be neither objectively evaluated nor standardized in terms of procedures. Furthermore, many times, face-to-face rehabilitation cannot be sufficiently frequent.

Telerehabilitation can cover the gaps for optimal continuity of care [5], and it can have equal or better effects compared to conventional face-to-face therapy due to the reproducibility of the tasks it uses [4]. When associated with assistive technologies, such as robots, virtual reality, tracking systems, or a combination of them, it could also allow the objective monitoring of patient progression [6]. Several hand-movement parameters that are important for rehabilitation can be automatically calculated and objectively evaluated and interpreted with the help of tracking systems [7]. Furthermore, these parameters can be also calculated for the healthy hand, for designing a useful standard, patient-specific, goal scale.

Other advantages of telerehabilitation are the possibility for patients to perform it at home, its reduced costs, and its integrability with other tools [8]. Indeed, an important role in telerehabilitation is assumed by the use of virtual, augmented, and mixed reality therapy (VAMR) or PC-based therapy [9], which has been proven to be very helpful [10,11,12]. Several works provided auxiliary systems for rehabilitation and/or telerehabilitation [13,14,15,16].

Telerehabilitation can be supervised and unsupervised [17]. In the first case, the patient cannot perform independent rehabilitation, being connected in real-time with the therapist to accomplish the task. This category mainly comprises wearable gloves (WGs) that are used to monitor the rehabilitation process [18,19,20,21,22]. It also includes the conventional face-to-face approach, in which the therapist supervises the procedure by dynamically modifying a subjective task, and the primary-secondary approach (PSA) method, in which the therapist, using a sensor-based glove (primary) transmits the motion data to a “secondary” glove worn by the patient that reproduces motion.

An example of the PSA is given by Mouri et al. [23]. This system includes a hand rehabilitation support system for the patient, an anthropomorphic robot hand for the therapist, and a remote monitoring system for determining the level of recovery. During the task, the therapist exerts force on the robot’s hand, and then the force is conveyed to the patient by the secondary support system. Another interesting implementation of the PSA is presented in [17], where both the therapist and the patient wear electromechanical gloves. Specifically, the therapist uses a sensor-based glove, whereas the patient uses a hand orthosis (a robotic exoskeleton) with pressure sensors. In [24], the authors presented an efficient method to acquire motion information from the primary unit while avoiding a physical glove, using instead an RGB-D camera with computer vision (CV) methods to track the therapist’s hand, and the patient still uses an electromechanical glove (EG). The primary unit tracks the 4D position of the therapist’s hand joints and remotely operates the exoskeleton. The system is also equipped with an external grip sensor, which measures the interaction forces and transmits this information to the therapist’s user interface.

Although these systems provide high precision, they have some important limitations: a robust and reliable Internet connection is required; the patient’s glove is wearable, so it can be expensive, cumbersome, limiting for hand movements, and not suitable for both hands; in the PSA, the power of VAMR/PC-based therapy is not exploited, due to the fact that the therapist controls the patient’s hand; the therapist’s presence and continuous participation are required through all the duration of the session.

In unsupervised rehabilitation, the treated person is able to perform the task independently, so it is not necessary to be constantly connected to the therapist. In this scenario, VAMR therapy and/or PC-based therapy are often used. Examples of this case are reported in [25], whose authors proposed a PC-based telerehabilitation solution for hand recovery which records EMG signals and heart rhythm during exercise. After each task, the former is used to provide the level of stress of the patient, whilst the latter provides information on the degree of muscle fatigue. However, no information about the joints’ positions is provided. Another PC-based hand telerehabilitation system is presented in [26], where the patient wears a WG, which records the positions of the hand joints while playing custom rehabilitation games on a commercial play station. However, in this system, the cost of the WG is very high. Instead, the authors in [27,28] showed a complete augmented reality rehabilitation system without WG but with a CV application for hand tracking using a 3D marker. Nonetheless, the positions of the hand joints are not precise.

Though the above systems can be very promising for supporting rehabilitation, most of them are also potentially employable for telerehabilitation. All of them are not focused on how a new (tele)-rehabilitation paradigm can be imagined when using these supporting instruments.

In this paper, we present a novel framework for hand telerehabilitation that allows cooperation between patients and therapists and includes a CV-based system named Virtual Glove (VG) [29,30,31,32,33] for real-time hand tracking (40 fps), which is transduced in a light, cheap, simple, and precise system to be used at home. The proposed architecture is designed to:support both supervised and unsupervised therapy;allow remote or “in-presence” therapy;support a client mode usage, robust in the case of a weak Internet connection;permit saving the history of the rehabilitation process of a person;provide an interface to the therapist for monitoring the rehabilitation progresses, managing patient and assigned rehabilitation tasks, data processing, and cooperating with other therapists by supporting data-sharing;recognize an administrator user that can manage the whole system.

## 2. The Cooperative Telerehabilitation Framework

In the following, we provide a description of the proposed telerehabilitation framework, whose ensemble is reported in Figure 1. The persons involved in the process, the actors, are the therapist, the patient, and the system administrator. From now on, we will mention these actors with capital first letters with respect to their roles in the system. The hardware consists of a VG and a server for long-term data storage and for hosting the software for data analysis and presentation. This last software consists of a GUI for assisting the Therapist in real-time interaction with the Patient, data extraction, processing, and presentation. A particular view of the same software will assist the System Administrator in managing the whole system and checking system’s security. The VG is located in the Patient’s house, while the server is located in a clinical environment. The Patient uses the VG directly, and the VG transmits the data to the server through an Internet connection. The Therapist and the Administrator can access the server remotely. In what follows, the hardware and the software architecture are described in detail.

### 2.1. Hardware Architecture and Data Model

Hand tracking is ensured by VG [33], a multisensor touchless device. The advantages of touchless systems are that they do not interfere with the hand’s free movement and are user-independent, meaning that they do not demand personalized equipment, or in the other case, equipment that many people should touch [16]. Since the system is driven by hand movements, patients with minimal finger mobility can also utilize it. The software is designed to be handled both by full-hand and by single finger movements, so patients can navigate through the procedure and practice exercises designed according to their capabilities. In the case of closed hands (for example, as a result of a stroke) the exercises can be modified to include minimal finger movements, something that is possible thanks to the spatial resolution of a few millimeters that the VG provides. The important advantage of using multiple sensors is overcoming occlusion issues, which have always complicated hand tracking. In the case where no object is handled, one single sensor cannot track the hand and finger locations with detail when the palm is turned perpendicularly to the sensor.

#### 2.1.1. VG Architecture

The VG was first presented in [29,30] using, then, a set of RGB sensors for hand tracking. In a new prototype [31,32], the RGB sensors were replaced with two infrared Leap Motion Controller (LMC) sensors [34,35,36] that, among other advantages, provide higher spatial resolution than visual light systems with a reduced effect of lighting conditions [37,38,39,40,41], and their own hand model that offers direct positioning of each hand joint, without the use of markers. The LMC is a small stereo-vision-based device for 3D visualization of the human hand’s movement. It includes three infrared (IR) LED sources and two IR cameras. It offers a spatial resolution of a few millimeters and frequencies that make it suitable for finger movement tracking [42].

Here, the system is composed of two LMC sensors mounted on rigid aluminum support used to fix the sensors in an orthogonal set-up. The proposed stereotactic, multisensor hand-tracking system needs calibration to align different views in a single-hand model. Calibration requires a precise two-step procedure to ensure precision and stability. In the first step, an external cloud of points was acquired by both sensors, and using singular value decomposition (SVD) analysis, rigid transformation matrices were acquired. For this step, the ability of the LMCs to detect not only fingers but also thin, pencil-like structures were used. A wooden stick was fixed to a numerically controlled mill, allowing it micrometric, three-axial movements. By moving the stick in the three axes along the region of interest, measurements of its position were acquired by both sensors. This way, a cloud of points was produced and used for the calculation of the transformation matrices.

In a second step, to ensure that internal differences between the two sensors were taken into account, measurements of the stick position were taken first by one and then by the other sensor, each put in the exact same positions. This resulted in calculating small axial translations that were integrated into the initial transformation matrices.

The resulting transformation is automatically performed in real-time on the acquired data, during the data collection from the two sensors.

The big advantage of a multisensor design is the possibility of measuring data from different orientations and reducing the occlusions. The disadvantage of using multiple LMC sensors is that they demand an equal number of operating systems, as one operating system cannot control more than one LMC, and even if recently it seems that this limitation is being overcome [35], it still remains for the oldest LMC devices. For this reason, the original architecture of the VG used a primary-secondary system configuration where each sensor was controlled by one of two virtual machines (Secondaries) hosted on a physical machine (Primary), where the data fusion also took place. In a second embodiment, taking advantage of the progress in the field of microprocessors, the PC hosting three servers was replaced by three Raspberry Pie (RP) machines [33], providing us with an economical and space-efficient solution. In this scheme, the raspberry Pie operating system is installed in every RP, and each LMC is associated with one machine. The third RP is where the data collection and fusion are performed, and it also hosts a virtual environment. In the RP version of our system, there was the constraint that the driver of the LMC needed to be run by an emulator of the X86 Machine on the Raspberry PI OS. However, due to the progress of the LMC drivers, by allowing those several LMC sensors to be managed by a single OS, the final deployment of the VG was done by using a single MINI PC Intel NUC with Intel core i5, 16 GB of RAM, and 256 GB of internal memory (Figure 2). The small size of the NUC gives us the advantage of portability and good performance.

By the described changes, we have created a system that for the first time does not demand the use of one or more personal computers but includes all devices needed, except from the screen, one set, that can also be contained (and sold) in one box, as a complete product.

#### 2.1.2. Storage System

As we mentioned before, it is important that the Patient can perform rehabilitation exercises from home or in a laboratory together with the Therapist. To ensure the performance of the telerehabilitation, we deployed the Server System in a PC with Intel I7, 32 Gb Ram, NVIDIA GE Force GTX 1080 to manage the large size of the data coming from the RS and TS. For the Therapist System and the Administrator System, the choice of the PC is not crucial as the software does not require great performance specifics. The connection between the macro-components was done via the WAN network. The location of the server is indicated in the upper-right part of Figure 1. For data storage, MongoDB was chosen as a tool to design the database. Unlike SQL databases, MongoDB does not use a fixed scheme for its documents. Since the data from a LMC is a JSON object, MongoDB is the best choice for this project because no preprocessing is necessary before inserting the data into the database; the collecting system maintains the data structure to facilitate data sharing among therapists or with third-party software. MongoDB is one of the fastest databases regarding reading/writing operations. This also allows streaming the Patient’s Task in real-time to the Therapist or extending the number of LMC used for the VG, with a reduced effort.

#### 2.1.3. The Data Model

LMC provides a skeletal hand model where all joints are represented. We use part of this model to obtain the hand skeletal model represented in Figure 3. In total, for each frame, the *X*, *Y*, and *Z* coordinates of 25 points are given, together with the respective time stamps (the temporal position of the current frame): 4 joints for the thumb; 5 from all other fingers; 1 is to represent the center of the palm. For the specificity of VG, information containing three hand models is saved on the dataset: one from the vertical LMC, one from the horizontal LMC, and one produced by their fusion. This way, for each time instant, a JSON object is saved containing the hand models. Additional information provided by the LMC, such as the orientation of the palm normal or a reliability score for each joint collected by a sensor, are also saved in the header of the streaming file, together with person-specific and task-specific information, for future exploitation.

In the VG, data fusion is performed in two possible ways. The first one is a binary switch approach. Here the palm’s normal direction (a vector orthogonal to the palm) defines what sensor contributes to the final result. More specifically, palm’s normal direction is utilised to compute the angle between the X-axis of the horizontal LEAP reference system and the projection of p on the X-Y plane. If the angle is between 225 and 315 (palm facing down) or 45 and 135 (palm facing up), the horizontal LEAP is activated, but the vertical sensor data is ignored. Outside this range, the sensors’ functions are reversed: vertical LEAP data becomes active, while horizontal LEAP data is disregarded (Figure 4).

A second method is a joint-wise switch approach where instead of taking data from the whole hand, from the same sensor in each frame, some joint is collected by a sensor and some are collected by the other. Here, the selection is done by comparing the inter-sensor with the intra-sensor velocity of each joint, based on the consideration that slower-tracked velocity probably signifies more robust data. Specifically, the metrics of intra- and inter-sensor joint speed are defined. If the inter-sensor spatial difference of the active sensor in consequent frames is higher than the intra-sensor one, this indicates tracking loss, and taking data from the other sensor is starting. This produces a final dataset where both sensors contribute to each frame [43].

For the sake of simplicity, in the embodiment described therein, just the first fusion method is used.

### 2.2. Use Cases and Software Architecture

The actors involved in the proposed framework are four: the Patient, the VG, the Therapist, and the Administrator (see Figure 5). The most important of those is the Patient, a person with hand mobility problems whose role is to perform rehabilitation tasks, as part of the rehabilitation procedure, using Rehabilitation System (RS) combined with the VG. The Patient should only focus on carefully completing his assigned tasks to the best of his ability, without having to take care of anything else.

When the RS turns on, the task personalized for the Patient is automatically loaded. When the task is completed, the RS saves the data externally in the Server System. This data-storage procedure is very important, since it allows the Patient to either run the task at home or in the presence of the Therapist, for a face-to-face rehabilitation session. The task has a double effect: motivation, needing a “competition” to reach a score; rehabilitation, driving the hand toward specific movements. The Therapist, once authenticated, can create, read, update, and delete (CRUD) regarding the Patient’s information, tasks, and session data.

Specifically, for the “create” operation: the Therapist inserts patient information into the system, and additionally, assigns to the Patient a physical machine (the VG). Once the “insert” operation is completed, the Therapist assigns a task. Additionally, the Therapist defines the duration, difficulty, and repetition frequency. The Therapist is able to monitor a remotely performed session in real-time, using streaming between the Therapist System and the RS. This is particularly useful the first time a treated person performs a new task or tries new settings of the same task (mostly in the case of a new difficulty level). A Patient–Machine–Therapist connection is generated, and the data files coming from that specific VG will contain, in its header, information about the Patient, the VG, the Therapist, and the executed task (type, difficulty level, score, etc.). A direct Therapist–Patient connection is also ensured by conventional channels, such as email and phone.

Once the task is complete and the data are stored from the RS on the Server System, the Therapist can load data and perform calculations and statistical analysis provided by the System through a specific GUI (detailed below). The Therapist can export a report on the resulting analysis and share it with colleagues for consultation and comparison. In the report, the Therapist can find the Patient’s information, together with selected plots.

The Administrator is responsible for the smooth operation of the whole system. He first appoints the Therapist and assigns to him the personal credentials to access the system. The Administrator is also responsible for standard data security, system maintenance, crash recovery, and software upgrades.

#### 2.2.1. Software Architecture

The system is divided into four macro-components (Figure 6): the RS, the Server System (SS), the Therapist System (TS), and the Administrator System (AS). RS executes the rehabilitation task. SS saves rehabilitation files. TS allows the Therapist to perform data analysis and visualization. Finally, AS allows the Administrator to supervise and manage the whole activity. In the following, these components are described in detail.

#### 2.2.2. Rehabilitation System

We created the RS to handle the rehabilitation process. The goal of this component is to let the Patient perform the tasks in a virtual reality (VR) environment by using the VG. It also locally saves the session data related to the task, and once the session is complete, it transfers the data to the server, inside the MongoDB database.

To collect data from both LMC of the VG, the LMC GEMINI 5.7.2 driver was used: this is the first one that can manage more than one LMC per operating system. This procedure allows the use of the UltraLeap Tracking Service (UTS) to collect the data from each LMC, build the numeric hand model, and share the corresponding model with the other components.

To use these data, we created the RehabilitationAPP component in the UNITY3D Game Engine. This application, using the data coming from the UTS, creates an VR environment with rehabilitation tasks. In detail, we integrated the Ultraleap Official Libraries (UOL) with the numeric hand models generated by the UTS. It is important to note that the hand data coming from the UTS are raw. The UOL applies a mirror transformation with respect to the *X*-axis, to present the virtual hand as seen from the point of view of the user (otherwise, a right hand would be seen as a left hand, and vice versa). Then, translation, rotation, and scaling are applied to the LMCs to locate them inside the Unity world’s space. In the same way, we also placed the two LMCs composing the VG inside the Unity world as they were located in the real world. The rigid transformation matrices derived from the calibration were applied to the LMCs in the Unity world: the translation and scales vectors as they were, and the rotation matrices in Euler angles. Afterward, since the UOL allows fusing the hand models by using the procedures reported above, for testing purposes we implemented the switching algorithm in which only one hand data model is chosen for each frame.

Finally, for testing purposes, we integrated the hand model into an example rehabilitation task consisting of a keyboard in which some buttons lit up randomly and had to be turned off with parts of the hand that also lit up.

Thus, when a rehabilitation session starts, RehabilitationApp loads on the screen-task-specific summary information, which is used to inform the Patient about the task he is trying execute. Then, by placing the hand on the Virtual Glove, RehabilitationApp loads the task with the settled task parameters. A ten-second counter starts, and when it finishes, the task starts; this allows the sensors to better track the hand for the execution. When the task time expires, RehabilitationApp checks if repetitions are necessary. If yes, RehabilitationApp loads a new screen with which the Patient can take a rest before putting his hand in the view of the VG to start the new repetition. If not, the system loads an end screen with the task execution summary.

All the real-time data generated by the VG during the rehabilitation session are saved in a support instance of MongoDB. Since Unity does not have an official integration of MongoDB, we created the LocalServer application to save the data. The RehabilitationApp and LocalServer are connected by a Websocket protocol. The support instance is integrated to avoid that Internet connection issues could degrade the system performance. Once the Patient finishes a task, the local server transfers the data from the support instance into the Server instance, from where it can be loaded and analyzed by the Therapist. Finally, we embodied the AnyDesk application to the RS to connect via streaming RS and TS.

#### 2.2.3. The Server System

For this project, three MongoDB collections were created named “User”, “Task”, “Task_Parameters”. Additionally, *n* collections are called “Session_unique_identifier”. The ”User” collection contains personal and identification information for each Patient or Therapist The collection “Task” stores objects regarding the 3D interactive tasks (name, description). The collection “Task_Parameters” contains information regarding configuration parameters of a specific task (duration, difficulty, repetitions, and starting date). Session_unique_identifier is the collection where the data from each rehabilitation session are stored. The unique_identifier is a string composed of id_userp (Patient), id_usert (Therapist), id_task_parameters, and current date. This way, each task session is saved in a unique collection with all necessary retrieval information on its name, and we can optimize the execution time for the queries by using a solution that only requires one collection for all the task sessions.

#### 2.2.4. The Therapist System

The TS was created with the aim of allowing therapists to administer services to their patients. To this end, TherapistAPP was developed. Its goal is to control in a simple and user-friendly way collections that contain actors and task data. It includes finding tasks by using as query criteria the Patient’s name, id, task type, or execution time, analyzing and visualizing the movements, and saving the results in an easy-to-read PDF form. In addition, the interface gives the possibility of comparing results from different sessions, so that the Therapist can monitor a therapy plan’s progress and the effect of the therapy plan on the mobility of the hand. At the same time, through the application, it is possible to create new structures and practice programs for new or already existing patients. Throughout the procedure, the original data remain intact, available for replicating the analysis with different criteria. The application is organized into two different tabs. One is used for the analysis of preexisting task data, and the second one gives the user the possibility of inserting new patients into the database and organizing their task plans.

Although the application was created in MATLAB, its code was converted into C++ using MATLAB Coder. In this macro-component, we integrated the AnyDesk application in order to ensure a streaming connection between the RS and the TS to let the Therapist monitor remotely the Patient during task execution.

#### 2.2.5. Administrator System

The AS is a MATLAB application, AdministratorApp, through which the Administrator is able to “create” a Therapist and allow him to manage data. Through the application, besides creating new Therapists, the Administrator can define/revoke the Therapist’s credentials. Furthermore, AdministratorApp allows the Administrator to edit information on Patient–Therapist links, allowing the Therapist assigned to a Patient to be changed if required. Additionally, in this case, MATLAB was used to define AdministratorApp, and then MATLAB Coder was used to generate the C++ code.

## 3. Execution Example

We imagine the execution of the pipeline inside a healthcare facility, and we describe it as reported in Figure 7. During a face-to-face meeting, the Therapist suggests the Patient uses the VG. Hence, the Patient is furnished with a VG, and at the same time, the Therapist requires the Administrator to have the credentials to assist in the rehabilitation through this framework. Once the Therapist has obtained the credentials, he is allowed to use the GUI. The Therapist uses these credentials for inserting the Patient into the system, associating him with the collected VG, and assigning him a task, through a GUI. The same credentials are used for accessing the database and performing data analysis. Making this procedure allows the VG to send all the data of the Task to the Server. Then, the Therapist teaches the Patient how to use the VG system with the assigned task. In Figure 8, the course of the data produced during a rehabilitation task is shown. The Patient turns on the VG, and the system loads the assigned task and executes it. Once the execution is completed, all data related to the task are moved to the central database in the Server System.

This way, from the GUI, the Therapist can interrogate the DB to load the required data, and analyze them by calculations—among others, hand statistics describing the position, speed, and acceleration of all individual finger joints. According to the procedure results and the Patient’s performance, the Therapist can then decide to assign new tasks or organize a new face-to-face meeting. Finally, the Therapist can export the Patient’s analysis data as a PDF file. This is important for sharing information with other therapists.

During the pipeline, the Administrator can make some modifications to the system—for instance, he can delete some old information regarding Patients, VGs, or Therapists.

## 4. Discussion and Conclusions

In this paper, we have presented the structure, both hardware and software, of a novel system designed for at-home telerehabilitation sessions. The novelty of this work lies in the fact that it is designed to give to the herapist quantitative and not only qualitative information about the patient’s mobility, for every hand joint separately, while at the same time providing the possibility of controlling the result of the rehabilitation plan by also quantitatively monitoring the progress of the hand’s mobility. The exercise plan takes the form of simple, easy-to-understand, and easy-to-perform tasks, creating less fatigue in comparison with exercise plans composed of repetitive movements, thereby making the task completion easier and more probable.

Although the main goal of this system is to be used for real rehabilitation purposes, it can also be used for research, since it provides information in an easy-access form, and it contains a database that can be used for sharing information between authorized people without risking the patient’s privacy. For this, it helps also that the hand movement is represented by a hand model; thus, not even visual information needs to be shared.

The computational systems that were used are small in size and relatively cheap, since we have replaced big personal computers with a MINI PC Intel NUC with Intel core i5, 16 GB of RAM, and 256 GB of internal memory, composing a system that can fit in a shoe box. It is economical enough to be a viable solution for patients that are in need of at-home rehabilitation but cannot afford the cost of multiple visits from a therapist. At the same time, being a personal device, it is safe for use by immunocompromised individuals, and suitable even in situations where self-isolation is needed.

A limitation of our work is that for now, the system has been preliminary tested only by healthy individuals 25 to 30 years of age with no diagnosed anatomical or functional hand issues. In these trials, the system has been found to be effective. More precisely, people that took the “Patient” role have mentioned that the interfaces are easy to perform, the guidelines of the tasks clear, and the exercise games work without issues. However, the system has not yet been tested by people with reduced hand mobility. In future work, trials by patients will be performed, and the collected data will be analyzed with advanced multi-variable strategies.

The goal of our system is also to provide the therapists with a useful tool for performing and evaluating telerehabilitation practices. In a future application, the therapists can use the quantitative data the system provides for validating the hand mobility in a standard goal scale. This is possible because VG can be used with both hands: a standard goal scale can be calculated by analyzing data from the healthy hand of the same patient, thereby making it possible to create a patient-specific goal scale. Furthermore, in future work, the functionality of the system, especially of the GUIs, needs to be evaluated by rehabilitation therapists in realistic approximations of or real therapy conditions, in order to identify possible points for improvement in all the parts of the architecture, including the possibility of constructing high-level interactive graphic user interfaces which allow the therapists to implement patient-specific, or better yet, infirmity-specific rehabilitation tasks, as in [44,45].

Right now, work is in progress in two directions: to provide a more effective strategy for hand-model fusion among different LMC, and to build a standardized data format as a new tool for data-sharing among therapists. This could also contain other-source multi-modal information (besides task-related information, it could also include, for example, simultaneous EEG recordings).

## Figures and Tables

**Figure 1 sensors-23-03463-f001:**
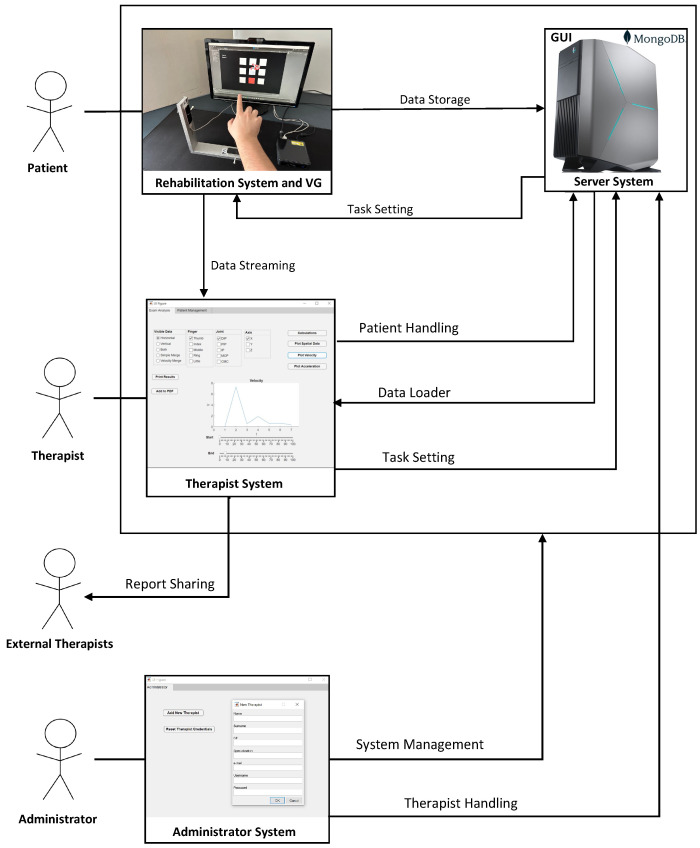
Hardware and software architectures of the framework. The hardware consists of the rehabilitation system with the VG and a server for data storage and for hosting the software. The software consists of a GUI for the Therapist and one for the Administrator. The persons involved in the process are the Therapist (or Therapists), the Patient, and the System Administrator.

**Figure 2 sensors-23-03463-f002:**
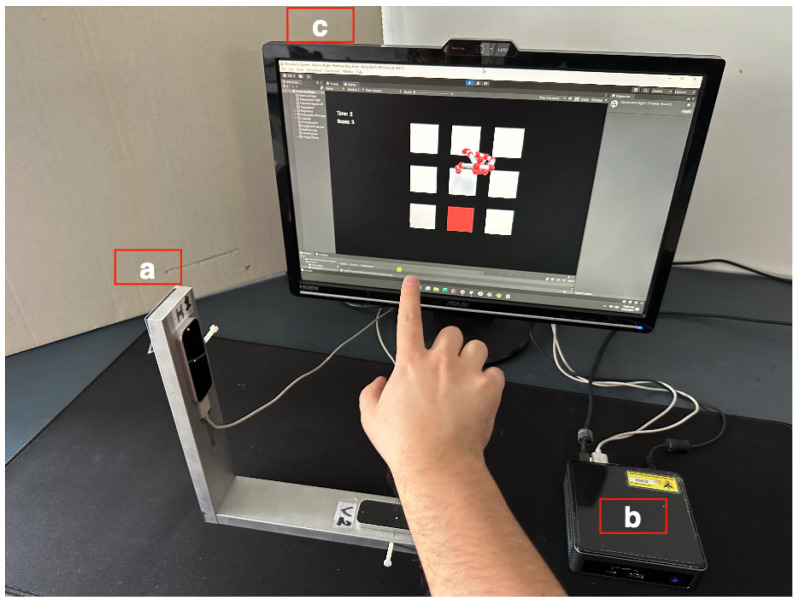
VG final implementation, including sensor assembly (**a**), an NUC-based driving unit (**b**), and a video screen for the active interaction (**c**).

**Figure 3 sensors-23-03463-f003:**
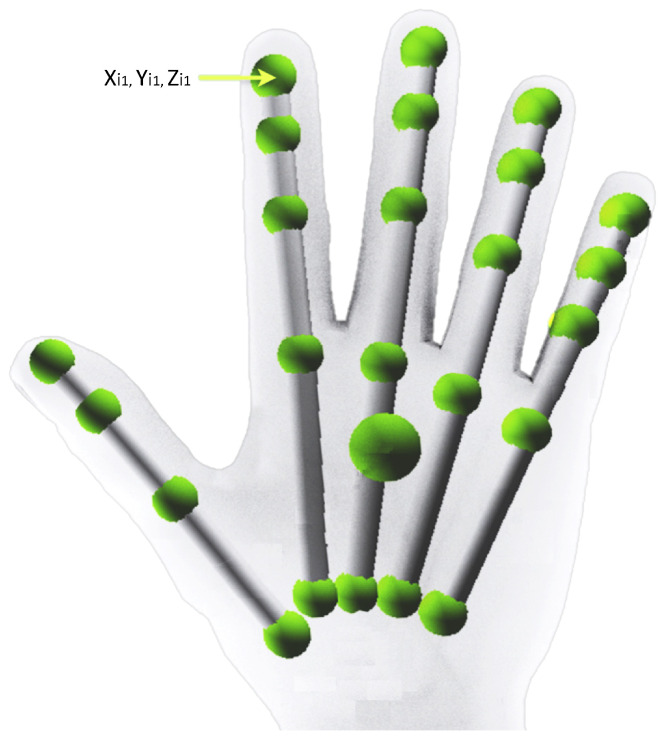
Skeletal hand model (skin contour is reproduced for reference) in which each joint is represented by a green sphere.

**Figure 4 sensors-23-03463-f004:**
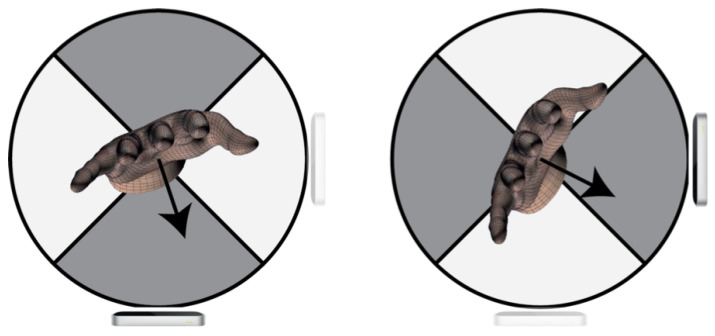
Binary switch approach: depending on the orientation of the hand, one sensor data are used, while the others are discarded; black arrows indicate the palm normal vector.

**Figure 5 sensors-23-03463-f005:**
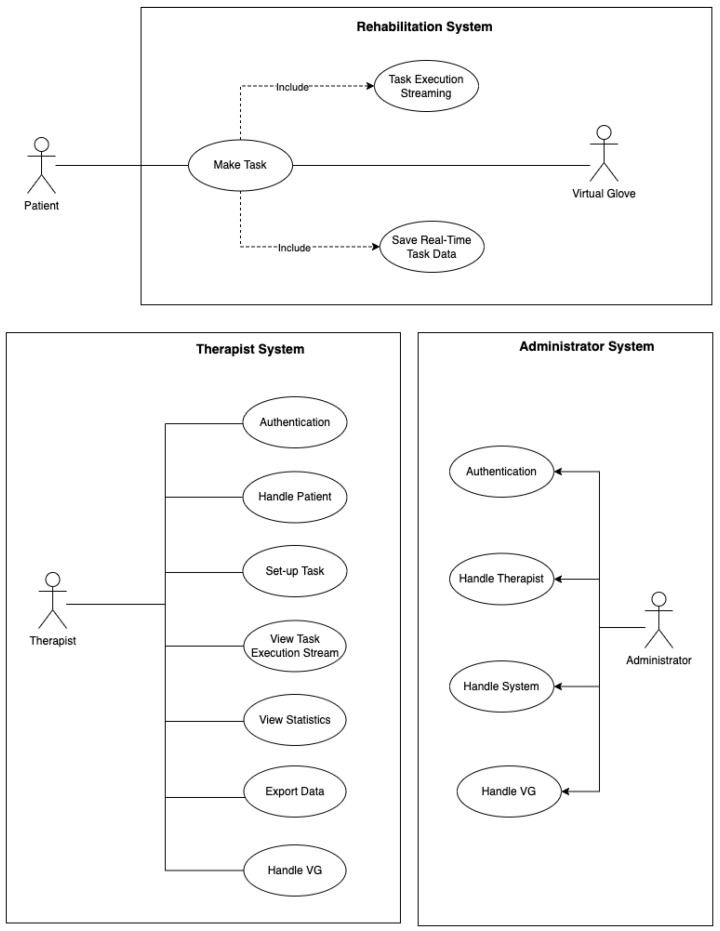
Use a case diagram presenting the functionalities and actors involved to the framework’s usage.

**Figure 6 sensors-23-03463-f006:**
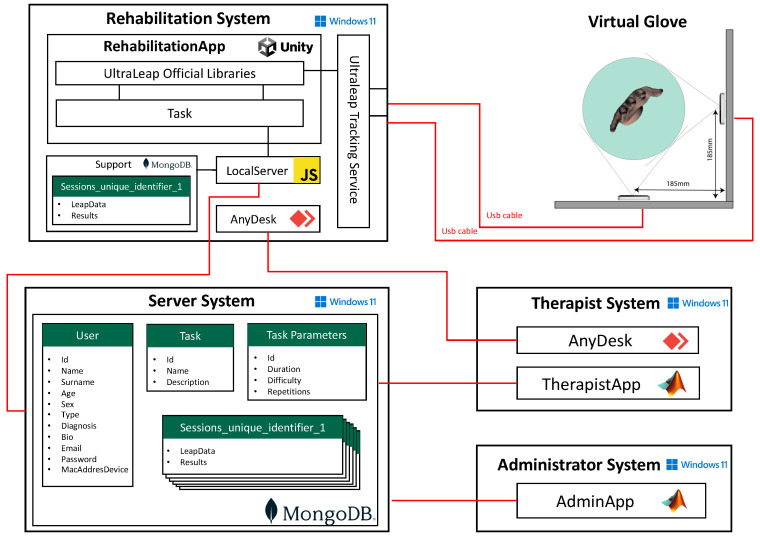
Software architecture diagram representing all the systems involved: Rehabilitation System, Server System, VG, Therapist System, and Administrator System.

**Figure 7 sensors-23-03463-f007:**
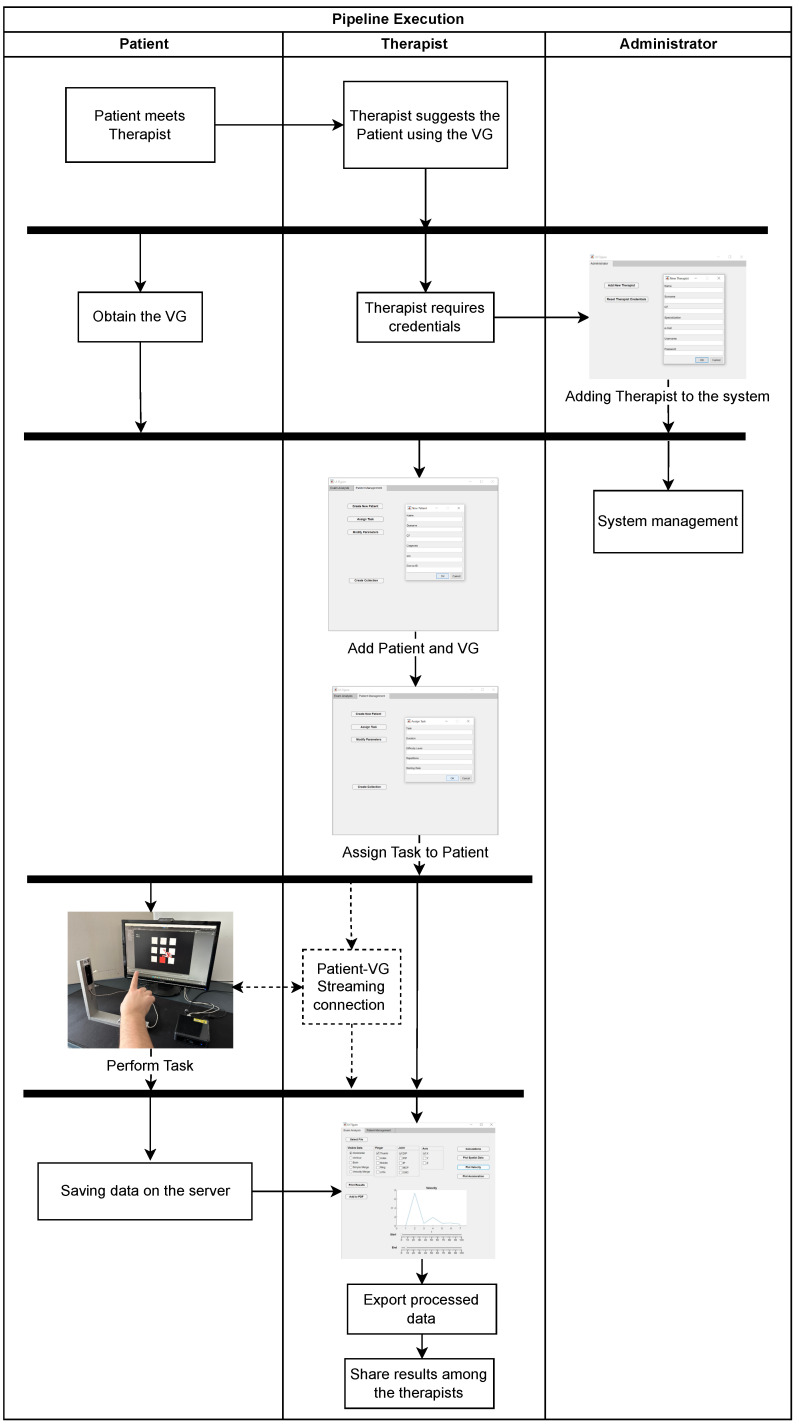
Swim lane diagram showing the pipeline’s execution. The pipeline describes the rehabilitation process, from the beginning when the Patient is suggested to use the VG, until data are processed and shared among therapists. Obviously, the process can be continued by the Therapist, assigning a new task or updating the task’s difficulty, or restarting with a new Patient.

**Figure 8 sensors-23-03463-f008:**
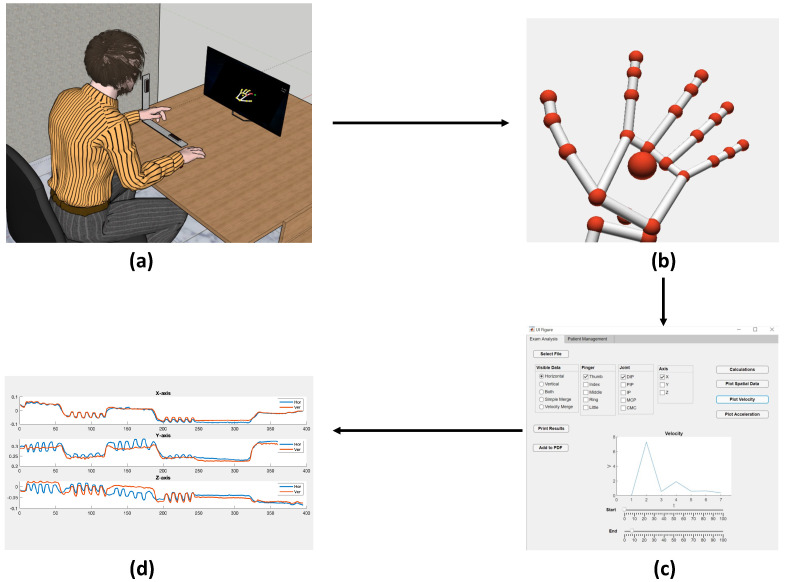
VG Data-flow pipeline. Here, one can see (**a**) the Patient performing a task, (**b**) the generated hand model, (**c**) the Therapist’s interface for data analysis and visualization, and (**d**) created plots presenting joint movement. The associated colors and the respective legend in (**d**) are irrelevant for the purpose of this paper: they just serve to show the general structure of the report.

## Data Availability

Data sharing not applicable.

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
