# Peer review of "Patient–Therapist Cooperative Hand Telerehabilitation through a Novel Framework Involving the Virtual Glove System"

_sensors, 2023, doi:10.3390/s23073463_

Round 1

Reviewer 1 Report

This research is very interesting, but will be improved using a correct multivariable analysis to understand better this implementation.

Is very important describe better this case of study, as in:

https://www.taylorfrancis.com/chapters/edit/10.1201/9781003191148-3/serious-game-caloric-burning-morbidly-obese-children-jos%C3%A9-d%C3%ADaz-rom%C3%A1n-alberto-ochoa-zezzatti-jose-mej%C3%ADa-mu%C3%B1oz-juan-cota-ruiz-erika-severeyn

In addition is important describe a Design of Experiments.

Using a more detail future research including the future limitations to this study as in:

Alejandro MoyaElena NavarroJavier JaenVíctor López-JaqueroRafael Capilla:
Exploiting variability in the design of genetic algorithms to generate telerehabilitation activities. Appl. Soft Comput. 117108441 (2022)

Is very important using a comparative table with this data associated with this research.

Author Response

I'm sorry but it was impossible to eliminate the previous author's cover letter by substituting it with the answer to the reviewers. By the way, I have provided it for the other reviewers (it is the same for all).  I apologize for that. Thank you

Reviewer 2 Report

Please find all my comments in the attachment.

Reviewer 3 Report

I found the article very interesting to read and novel, what I recommend is to specify more how it would be used in the case of closed hands in stroke patients since placing the sensors in this type of hands would be very complicated and is not fully described in the text. And it could also be included if any assessment scale of the upper limb has been taken into account in order to validate the data with a standard goal scale, and thus be more reliable for the rehabilitators who may use the device. 

Reviewer 4 Report

The article presents a new telerehabilitation system that can be used by the patient at home both with and without direct supervision by a specialist. Also, with the help of this system, reports on completed tasks can be received and the possibility to share these results with other specialists has been implemented. The advantage of the system is that there is no need to place markers that would limit the patient's movements during the performance of the tasks. Implementation of such a system in clinical practice would greatly improve the work of rehabilitation specialists.

The work is very interesting and valuable, but I have some questions and comments:

1. The description of your proposed system is really very detailed. However, in reviewing your work, I missed an important piece of information that I believe is extremely important: What would be the useful parameters that I, as a rehabilitation professional, would gain from using this system? Is it possible to measure both arms at the same time? Does the specialist have to interpret the results himself, or are algorithms implemented in the system to facilitate this process? Due to the lack of time in clinical practice, the rapid acquisition and interpretation of results is very important, as rehabilitation specialists usually avoid systems that are complex and require additional knowledge to make a final decision.

2. In the discussion, I would like a more detailed comparison with existing similar systems and a description of limitations.

Minor comments:

• The WG abbreviation is already explained in line 40, so there is no need to repeat it in line 74.

• In line 77 should the AR abbreviation explained. 

• I would suggest moving most of the title text of Fig.1 to Chapter 2. 

• Line 114 “The LMC”?

Reviewer 5 Report

The article describes a hand telerehabilitation system for post-stroke survivors using a virtual glove. The authors explained the system in details; but I was also expecting a performance evaluation of the telerehabilitation system and/or its usability evaluation from stakeholder’s perspective. Also, I was unclear about the difference with its previous research publications in references 26-30. I’d recommend to reduce the number of self-citations, particularly if they are improvements of previous designs. For example, ref 27 is the improvement of ref 26; same for ref 30 and 28.  Also, explain how this current article is different from its latest publication. These comments among minor comments below should be addressed to be considered for publication. Minor comments below:

1.      Line 63. It wasn’t clear how the VAMR can be used in telerehabilitation and why it would be helpful with the MSA approach. Please explain

2.      Line 86. Should address cost and the use VAMR as part of innovation. How is the latter used in the proposed device?

3.      Figure 1 caption should be provided within the text.

4.      Line 110, 148. Design improvements seemed incremental. Provide major innovation of the proposed system compared to currently published.

5.      Line 43: I’d recommend to use alternative language to describe master/slave (e. g. primary/secondary)

6.      Line 114. Typo?

7.      Line 167. Provide sampling time to address its ‘real-time’ process.

8.      Main concern is how the quantitative data can be standardized to reduce subjective assessment. This should be further explored to translate the research from laboratory into clinical practice. Maybe provide it as future work if not addressed already.

9.      Line 190-195. Repeated paragraph. Delete it.

10.   Line 236. What measures does the Therapist receive in a report? How did the research team decide the priority or importance of these measures?

11.   Other technical questions: How far does the hand need to be located from the sensors? How does the user maintain the position of the hand and possible offset orientation from sensors? Typically, post-stroke survivors require some physical assistance, whether from therapist or an assistive technology, during early rehabilitation. How does the hand telerehabilitation system reduces the need of this type of assistance?

Round 2

Reviewer 5 Report

The authors addressed the reviewers' comments.